# Profiling of Saharan dust from the Caribbean to western Africa – Part 2: Shipborne lidar measurements versus forecasts

Albert Ansmann[1], Franziska Rittmeister[1], Ronny Engelmann[1], Sara Basart[2], Oriol Jorba[2], Christos Spyrou[3], Samuel Remy[4], Annett Skupin[1], Holger Baars[1], Patric Seifert[1], Fabian Senf[1], and Thomas Kanitz[5]

[1]Leibniz Institute for Tropospheric Research, Leipzig, Germany
[2]Barcelona Supercomputing Center, Dep. of Earth Sciences, Barcelona, Spain
[3]National and Kapodistrian University of Athens, Dep. of Physics, Athens, Greece
[4]Laboratoire de Météorologie Dynamique, IPSL, UPMC/CNRS, Paris, France
[5]ESTEC, Noordwijk, The Netherlands

*Correspondence to:* A. Ansmann
(albert@tropos.de)

**Abstract.** A unique 4-week ship cruise from Guadeloupe to Cabo Verde in April-May 2013 (see part 1, Rittmeister et al., 2017) is used for an in-depth comparison of dust profiles observed with a polarization/Raman lidar aboard the German research vessel Meteor over the remote tropical Atlantic and respective dust forecasts of a regional (SKIRON) and two global atmospheric (dust) transport models (NMMB/BSC-Dust, MACC/CAMS). New options of model-observation comparisons are presented. We analyze how well the modeled fine dust (submicrometer particles) and coarse dust contributions to light extinction and mass concentration match respective lidar observations, and to what extent models, adjusted to aerosol optical thickness observations, are able to reproduce the observed layering and mixing of dust and non-dust (mostly marine) aerosol components over the remote tropical Atlantic. Based on the coherent set of dust profiles at well defined distances from Africa (without any disturbance by anthropogenic aerosol sources over the ocean) we investigate how accurately the models handle dust removal at distances of 1500 km to more than 5000 km west of the Saharan dust source regions. It was found that (a) dust predictions are of acceptable quality for the first several days after dust emission up to 2000 km west of the African continent, (b) the removal of dust from the atmosphere is too strong for large transport paths in the global models, and (c) the simulated fine-to-coarse dust ratio (in terms of mass concentration and light extinction) is too high in the models compared to the observations. This deviation occurs initially close to the dust sources and then increases with distance from Africa, and thus points to an overestimation of fine dust emission in the models.

## 1 Introduction

In a companion paper (Rittmeister et al., 2017), we present Saharan dust observations over the remote tropical Atlantic Ocean between Guadeloupe (16° N, 61° W) and Cape Verde (17° N, 25° W). Dust profiles were continuously measured during a 4–week cruise with a polarization/Raman lidar aboard the German research vessel (R/V) Meteor. The transatlantic cruise over 4500 km is shown in Fig. 1 and took place from 29 April to 23 May 2013. The lidar observations were conducted in the frame-

work of the Saharan Aerosol Long-range Transport and Aerosol-Cloud Interaction Experiment (SALTRACE) (Weinzierl et al., 2017). Dust profile observations along the main outflow route of Saharan dust towards North America with a slowly moving ship provide a unique opportunity to check the quality of forecasts of dust predictions models, especially the dust removal parameterization. No additional continental aerosol source disturbed the lidar observations over the remote tropical Atlantic, no orographic effect influences the air flow, and frontal activity and associated large-scale lifting of air masses causing complex vertical aerosol layering is absent over the tropical Atlantic.

There is a strong need for the evaluation of dust transport models. Huneeus et al. (2011, 2016) provide an extended overview of the status of dust transport modeling and forecasting. Since mineral dust is ubiquitous and thus influences weather and climate, horizontal visibility, air quality, and human health with extreme effects during strong dust outbreaks, precise forecasting of dust events is a major issue for environmental and meteorological services and, more general, for atmospheric sciences. Significant progress has been made in dust modeling during the last ten years and a suite of regional and global dust transport models is available.

Aerosol Robotic Network (AERONET) sun photometer observations (Holben et al., 1998) play a strong role in dust model evaluation (see recent studies, e.g., of Scanza et al., 2015; Cuevas et. al., 2015; Ridley et al., 2016) because of the continuity of observations since decades, the easy and free access to the data, and high quality of AERONET products. However, dust column information as provided by AERONET observations is not sufficient. The life time and spread of dust on regional to intercontinental scales sensitively depends on the height at which dust is transported and also on the dust size distribution, and thus on fine and coarse dust fractions during emission and long-range travel. Per definition, particles with diameters $<1$ $\mu$m belong to the fine dust fraction and coarse-mode particles have diameters $>1$ $\mu$m. Dust removal (wet deposition and wash out, dry deposition by gravitational settling and turbulent downward mixing) is a strong function of height. Gravitational settling also sensitively depends on the simulated fine and coarse dust fractions. The same holds for dust radiative effects which are rather different for fine and coarse dust particles (Nabat et al., 2012; Ridley et al., 2016; Kok et al., 2017). Meanwhile, dust simulations are also used to estimate dust ice-nucleating particle (INP) concentrations (Hande et al., 2015; Nickovic et al., 2016). Dust cloud condensation nucleus (CCN) concentrations can be estimated as well (Mamouri and Ansmann, 2016; Hande et al., 2016). The CCN and INP number concentrations also depend on the simulated fine and coarse dust particle fractions.

Consequently, lidar observations are required for an improved evaluation of dust prediction models (Koffi et al., 2012, 2016; Mona et al., 2014; Cuevas et. al., 2015; Binietoglou et al., 2015). Polarization lidars which allow us to separate dust from non-dust aerosol profiles in terms of light extinction coefficient and mass concentration are most useful in this respect. Recently, it was shown that even fine dust and coarse dust can be distinguished (Mamouri and Ansmann, 2014, 2017) so that one of the most important modeling aspects which deals with the emitted and transported dust particle size spectrum (Kok, 2011a, b; Kok et al., 2017) can now be illuminated in model-observation comparisons.

Our study is guided by the following main question: How well can state-of-the-art dust (regional/global) forecast models (SKIRON, MACC/CAMS, NMMB/BSC-Dust, see Sect. 3) reproduce our shipborne lidar observations of light extinction and mass concentration profiles as a function of transport length from about 1500 km to more than 5000 km from the dust source regions? A special focus is on dust removal and the question to what extend the simulated fine and coarse dust fractions are in

consistency with the lidar observations? In this context, we discuss the hypothesis of Kok (2011a, b) and Nabat et al. (2012) that dust models overestimate the fine dust fraction and underestimate the coarse dust fraction.

The cruise with a state-of-the-art continuously running dust profiling lidar across the Atlantic and the comparisons with the dust forecasts gives us, in addition, the favorable opportunity to inform the broader atmospheric science community about the recent progress in dust lidar observations and data analysis methods. We present the full set of products of the recently introduced POLIPHON (Polarization Lidar Photometer Networking) method (Mamouri and Ansmann, 2014, 2017).

The paper is structured as follows: In Sect. 2, details to the ship cruise as part of our lidar networking activities are given. The POLIPHON data analysis is explained and applied to four observations of key stages of dust layering over the tropical Atlantic (see Fig. 1, cases 1-4). These four cases were introduced and discussed by Rittmeister et al. (2017). The dust models are then briefly described in Sect. 3. The lidar observations of the four cases are compared with the model results in Sect. 4. The performance of the three dust models is discussed. The new options of comparison are applied.

## 2 R/V Meteor cruise as part of PollyNET and lidar data analysis

The continuously operated multiwavelength polarization/Raman lidar Polly (*PO*rtab*L*le *L*idar s*Y*stem) (Engelmann et al., 2016), used in our study for dust profiling during the transatlantic cruise M96 of the German R/V Meteor, is the key instrument of the so-called OCEANET-Atmosphere platform and is installed inside a container. The OCEANET-Atmosphere platform is usually operated during north-south cruises of the German ice breaker R/V Polarstern between Bremerhaven, Germany, and Cape Town, South Africa, or Punta Arenas, Chile in spring and autumn of each year (Kanitz et al., 2011, 2013) and thus routinely observes Saharan and Patagonian dust outbreaks. The OCEANET Polly lidar is the mobile platform of the lidar network PollyNET (Baars et al., 2016) which consists of permanent and temporary continuously running automated multiwavelength lidar stations. PollyNET is part of the European Aerosol Research Lidar Network (Pappalardo et al., 2014). One of the goals of PollyNET is the built up of a small network of autonomously running dust monitoring lidar stations. Polly stations close to deserts are available in Cabo Verde, Portugal, Greece, Cyprus, Israel, and Tajikistan. The aim is to support dust forecast modeling and to pave the way towards lidar data assimilation into dust forecast models.

The Polly data analysis software was extended during recent years by introducing the POLIPHON retrieval technique which is applicable to polarization lidar observations (Mamouri and Ansmann, 2014, 2017). Polarization lidar is a very powerful remote sensing tool for aerosol and cloud research. The technique permits the discrimination of desert dust or volcanic dust from other aerosols such as biomass-burning smoke, maritime particles, or urban haze by using the measured profiles of the particle backscatter coefficient and linear depolarization ratio (Sugimoto et al., 2003; Shimizu et al., 2004; Nishizawa et al., 2007; Tesche et al., 2009; Ansmann et al., 2012). The recently introduced POLIPHON approach enables us not only to decompose the measured aerosol profiles in dust and non-dust aerosol components in terms of particle extinction coefficient and mass concentration, but also to separate and estimate vertical profiles of fine dust and coarse dust light extinction and mass. These new features provide the unique opportunity in forecast-vs-observation studies to illuminate even the simulated size distribution characteristics in dust transport models as a function of transport distance from the source region, as performed in Sect. 4. In

this section, we apply the POLIPHON method to the four measurement cases we will use in our forecast-vs-observation study in Sect. 4 and explain the method and retrieval products in detail. The polarization lidar method is robust and simple, and does

not need any sophisticated particle shape model for the irregularly shaped dust particles in the data analysis.

The retrieval of profiles of the basic lidar products, i.e., of the particle backscatter coefficient, extinction coefficient, lidar ratio, and linear depolarization ratio from the Polly observations is described by Engelmann et al. (2016) and Baars et al. (2016). The profiles of the 532 nm particle backscatter coefficient and the depolarization ratio are input in the POLIPHON method. The basic particle optical properties of the four cases studied here are presented in the companion paper of Rittmeister et al.

10   (2017).

The POLIPHON retrieval consists of four steps. These steps are shown in Fig. 2 (c-e). We applied the method to case 4 in Fig. 1. On 5-6 May 2013, the R/V Meteor was 600 km west of Barbados. An aged Saharan dust plume was detected between 800 and 2000 m height after 9 days of travel across the tropical Atlantic. As mentioned, the height profiles of the particle backscatter coefficient (Fig. 2a) and particle linear depolarization ratio at 532 nm (Fig. 2b) as presented and discussed in

Rittmeister et al. (2017) are input of the POLIPHON data analysis, as mentioned.

In the first step, the vertical profiles of the particle depolarization ratio and backscatter coefficient are used to separated the dust and non-dust contributions to the measured (total) particle backscatter coefficient (Fig. 2c). Non-dust aerosol components are biomass burning smoke, anthropogenic haze (from industry, traffic, heating), rural (background) aerosol, and marine particles. A depolarization ratio of >0.31 and <0.05 indicates pure dust and pure non-dust aerosol, respectively. Depolarization

ratios from 0.05–0.31 indicate mixtures of dust and non-dust aerosol and can be easily quantified by applying the classical separation technique (Sugimoto et al., 2003; Shimizu et al., 2004; Tesche et al., 2009). This basic separation technique is denoted as one-step method of the POLIPHON retrieval scheme (Mamouri and Ansmann, 2014). The solutions are shown in Fig. 2c for the dust (red) and non-dust (green) particle backscatter coefficient.

In the next step (step 2), we convert the backscatter coefficients into light-extinction coefficients. In our specific case of

shipborne measurements west of Africa, we multiply the dust backscatter values with a typical extinction-to-backscatter ratio (lidar ratio) of 55 sr for western Saharan dust (Groß et al., 2011; Tesche et al., 2011; Haarig et al., 2017a) to obtain the height profiles of the dust extinction coefficient at 532 nm. According to our extended pure dust lidar-ratio observations in Morocco, Cabo Verde, and Barbados during the summer half years of 2006, 2008, 2013, and 2014 (Tesche et al., 2011; Haarig et al., 2017a) the uncertainty in the used dust lidar ratio of 55 sr is about 10 sr, and thus the relative uncertainty in the derived profiles

of the 532 nm extinction coefficient caused by the assumed dust lidar ratio is 20%. The result of the extinction retrieval is shown in Fig. 2d. The non-dust backscatter coefficients are converted to extinction coefficients as well by using typical non-dust lidar ratios. To cover the entire spectrum of possible non-dust aerosol scenarios from pure marine to pure continental aerosol, we calculate marine particle extinction coefficients by using a typical marine lidar ratio of 20 sr in the conversion (Groß et al., 2011; Rittmeister et al., 2017) and a continental (non-dust) aerosol extinction coefficient by taking a typical lidar ratio of 50 sr

for anthropogenic aerosols at 532 nm (Müller et al., 2007a). All three extinction profiles (dust in red, marine in blue, and anthropogenic in green) are presented in Fig. 2d in this way. Error bars show the estimated overall uncertainty resulting from

the basic particle backscatter retrieval, the dust and non-dust backscatter separation, and the lidar ratio estimate. More details to uncertainties can be found in Mamouri and Ansmann (2017).

Afterwards (step 3), we check, whether the sum of marine and dust extinction profiles or the sum of anthropogenic aerosol and dust extinction profiles approximately agrees with the particle extinction profile at 532 nm as independently obtained with the Raman-lidar method (see Mamouri and Ansmann, 2017, for more details). As can be seen in Fig. 2d, the Raman lidar solution for the total particle extinction coefficient (black curve) is already close to the dust extinction profile in the Saharan air layer (SAL) and close to the marine extinction coefficient in the marine aerosol layer (MAL) below the SAL. Raman lidar solution for the extinction coefficients are determined from nitrogen Raman backscatter profiles and uncertainties in the correction of the laser-beam to receiver-field-of-view overlap correction prevent a trustworthy determination of the extinction profile at heights below 600 m. All other lidar profiles shown in Fig. 2 (a-e) are trustworthy almost down to heights close to the surface because they are calculated from lidar signal ratios so that the overlap impact widely cancels out.

As outlined in detail in Rittmeister et al. (2017), the identification of aged smoke in the SAL after long-range transport over more than 5000 km within 9–10 day is complicated because fire smoke particles may significantly grow by water uptake during long-range transport and, as a consequence, change their optical properties (Müller et al., 2007b). The lidar ratio of fresh smoke may be as high as 50–70 sr, but can decrease significantly, e.g., to values around 30 sr (as is the case here) after such a long travel over more than a week. Thus, it is not clear in Fig. 2d whether the small remaining non-dust contribution to light extinction in the SAL is caused by marine or aged smoke particles or a mixture of both.

We repeated the procedure of the identification of the non-dust aerosol component for the other three cases. The results are shown in Fig. 3. Again, the Raman-lidar profile of the total particle extinction coefficient is already close to the dust extinction profile in the SAL in cases 1 and 3, and indicates pure marine aerosol conditions in the marine layer below the SAL in case 3. The weak non-dust aerosol contribution to total light extinction in the SAL (case 3) may indicate the presence of marine particle (lidar ratio of 20sr) or of aged smoke (lidar ratio of 30 sr). In case 2, we clearly observed a mixture of dust and smoke and other anthropogenic aerosol particles (causing a lidar ratio of 50 sr or an even slightly higher value) in the SAL at heights >1.8 km. The particle extinction profile obtained from the Raman lidar measurement and the dust extinction profile are only close to each other for the lowest part of the SAL. The Raman-lidar extinction values are again close to the marine extinction coefficients in the MAL in case 2. In the model-observation comparison in Sect. 4.1, we ignore the smoke and haze impact in the SAL (case 2) for simplicity, and concentrate on marine and dust particle extinction and mass concentration profiles. Marine particles dominated in the MAL and desert dust in the SAL.

In the final step (see Fig. 2e), we convert the extinction coefficients for dust, marine, and anthropogenic aerosol into respective mass concentration values by using appropriate extinction-to-volume conversion factors given in Mamouri and Ansmann (2017). The respective conversion factors were obtained from multi-year AERONET sun photometer observations and from field campaigns with lidar/photometer combinations. An extended discussion of the uncertainty in the applied conversion factors is given in Mamouri and Ansmann (2014). In the conversion of volume concentrations into mass concentrations we assume particle densities of 1.2 g cm$^{-3}$ (marine), 1.55 g cm$^{-3}$ (continental pollution) and 2.6 g cm$^{-3}$ (dust). As shown in Mamouri and Ansmann (2014), the overall uncertainty in the estimated dust mass concentrations is of the order of 25–35%.

A highlight of the article is the comparison of fine dust and coarse dust extinction and mass concentration profiles. By means of a new option, denoted as 2-step method in the POLIPHON retrieval framework, we can separate fine dust and coarse dust profiles in terms of 532 nm light-extinction and mass concentration. It is however out of the scope of this article to explain the full methodology. Details can be found in Mamouri and Ansmann (2017). The technique makes use of the fact that fine dust and coarse dust show different depolarization ratios of 0.14–0.18 (fine dust, 532 nm) and 0.35-0.39 (coarse dust, 532 nm). We adjusted the coarse-mode depolarization ratio to 0.35 for aged dust when the very large dust particles are removed after long-range transport. The fine and coarse dust backscatter, extinction, and mass concentration profiles for the measurement example in Fig. 2 are shown in Fig. 4. The lidar method assumes vertically homogeneous dust properties within the SAL and thus a height-independent dust size distribution so that the ratio of fine dust to coarse dust extinction coefficient and mass concentration is also height independent. This assumption is fully justified for aged dust layers over the Atlantic as the almost height-independent lidar profiles of the dust depolarization ratio and lidar ratio during the SALTRACE summer campaigns in 2013 and 2014 indicate (Haarig et al., 2017a). The dust depolarization ratio and lidar ratio would sensitively change with height when the size distribution characteristics would significantly change from Sahara dust layer base to top.

The obtained fine dust and coarse dust profiles of our cases 1-4 are in full agreement with the shipborne sun photometer observations of fine dust and coarse dust AOT at 500 nm on the four days, as will be shown and discussed in more detail in Sect. 4. During the SALTRACE campaign in June-July 2013, we compared the lidar-derived fine-dust volume and mass concentration profiles with respective near-by airborne in situ dust observations and found very good agreement within 20%. The difference is a result of atmospheric variability and uncertainty in the aircraft and lidar observations.

The uncertainties in the lidar products are indicated by error bars in Figs. 2 to 4. These error bars include the uncertainty in the separation of dust and non-dust aerosol components, in the assumed lidar ratios in the retrieval of the extinction coefficients, and in the extinction-to-mass conversion factors. Typical overall uncertainties in all presented parameters are discussed in Mamouri and Ansmann (2017).

## 3 Atmospheric modeling systems

In this section, we provide an overview of the three dust forecasting models used in our study. Information on the main model characteristics is given. Differences and common features regarding the modeling of dust emission, transport, and deposition are discussed. Proper modeling of emitted dust, the size distribution during emission and processes that influence the dust size distribution during long-distance transport is essential for an accurate simulation of the dust life cycle. How well the model parameterizations work, will be studied in Sect. 4.

### 3.1 SKIRON

The SKIRON modeling system is an integrated limited area modeling system (a regional dust prediction systems) developed at the National and Kapodistrian University of Athens (Kallos et al., 2007; Astitha et al., 2008; Spyrou et al., 2010) in the framework of the nationally and European Union (EU) funded projects SKIRON, Mediterranean Dust Experiment (MEDUSE),

Atmospheric Deposition and Impact on the Open Mediterranean Sea (ADIOS), and recently Climate Change and Impact Research (CIRCE) and Marine Renewable Integrated Application Platform (MARINA). The SKIRON system simulates the dust cycle (uptake, transport, deposition and its impacts on radiation) and provides dust load and deposition forecasts (available at http://forecast.uoa.gr/dustindx.php). The name SKIRON (one of the eight gods of winds) is taken from the Greek mythology.

The SKIRON model covers the entire area of Northern Africa, the Mediterranean Sea, Europe, the North Atlantic Ocean and part of the East Coast of the United States and the Caribbean. This particular setup runs operationally at the servers of the National and Kapodistrian University of Athens and has a horizontal resolution of $0.2° \times 0.2°$. Only dust, no other aerosol component, is considered in the model. On the vertical it uses 38 levels from the surface to the top of the atmosphere. The main goal of this setup was to examine the trans-Atlantic transport of Saharan dust and associated weather patterns; therefore such a substantial domain is needed. The outputs from this operational model are the ones used in this manuscript. No weather data assimilation is performed (apart from the analysis fields used every day for initial conditions) due to the size of the area covered. It is difficult to have daily observations, which are not included in the analysis fields used for initialization, especially for the Atlantic. The forecast range evaluated is 24 hours, i.e., the first forecast day.

A contrasting feature to the other two models used here is the formulation of the source function (characterization of accumulated sediments, soil texture, porosity, bulk density, composition, soil moisture, and soil particle size distribution). SKIRON uses the soil texture database developed by Miller and White (1998) which, in turn, is based on the US Department of Agriculture's State Soil Geographic Database. The atmospheric model is based on the ETA/National Centers for Environmental Prediction (NCEP) model but is heavily modified to include state-of-the-art parameterization schemes for meteorological and desert dust processes (see Spyrou et al., 2010, for details). Recently the model was updated to use the RRTMG (Rapid radiative transfer model for global circulation models) radiative transfer scheme (Iacono et al., 2008) in order to calculate the radiative feedback of dust particles in the atmosphere (Spyrou et al., 2013).

SKIRON uses a modal representation of the particle size for a more accurate description of the aerosol mass distribution over the source areas, as well as for the description of long-range transported dust particles. More specifically, the dust particle size distribution follows a lognormal form with mass median diameter equal to 2.524 $\mu$mm and geometric standard deviation equal to 2. Currently, the transport mode uses eight size bins with effective radii of 0.15, 0.25, 0.45, 0.78, 1.3, 2.2, 3.8, and 7.1 $\mu$m. Table 1 shows the size bins (radius classes) used in SKIRON and for comparison the particle radius classes implemented in the other two models described below. The dust scheme (emission, transport, dry and wet deposition features) is described in Spyrou et al. (2010). Fine and coarse dust fractions are set constant in the model, and therefore do not change during dust transport. The mass-related fine mode fraction (FMF) is about 0.07–0.1 in the model throughout the entire dust life cycle. Dust removal parameterizations are similar in all three models and described in more detail in Sect. 3.3.

The simulated dust mass concentration profiles were provided by the University of Athens (from the operational dust forecasting system) for the nearest model grid points to and around the R/V Meteor and the nearest time step to the lidar observations (cases 1–4). We computed the dust mass concentration profiles for each ship position and for 8 points around the research vessel and calculated the average dust profile.

The particle extinction coefficients for 550 nm were then calculated from the simulated size-class-resolved mass concentrations by using the effective radius per size class representing the standard size of the particles of each of the eight classes, the dust density of 2.6 g cm$^{-3}$, and literature values of dust refractive indices and dust extinction efficiencies for the given eight effective radii. Spherical dust particles are assumed in the calculation of the extinction coefficients. A detailed description of the computations of dust extinction coefficients from the modeled dust mass concentrations is given by Tegen and Lacis (1996), Pérez et al. (2006), and Spyrou et al. (2013).

## 3.2 MACC/CAMS

MACC (Monitoring Atmospheric Composition and Climate) was developed within the framework of an European Union (EU) project (*http://www.gmes-atmosphere.eu/about/project/*) which was coordinated by ECMWF (European Centre for Medium-range Weather Forecasts, Reading, United Kingdom). MACC activities are now carried on under the Copernicus Atmosphere Monitoring Service (CAMS). The simulation system MACC/CAMS combines state-of-the-art atmospheric modeling with **operationally assimilated** Earth observation data to provide **near real time (NRT) forecasts for a wide range products regarding** European air quality, global atmospheric composition, climate forcing, the ozone layer and UV, solar energy, and emissions and surface fluxes. The MACC/CAMS simulation system gets input data from satellites, in situ measurements and information about aerosol emissions and fires. The anthropogenic emissions are based on established inventories. Two main global MACC/CAMS products are analyses and forecasts of aerosols (*http://www.gmes-atmosphere.eu/*). Details to this complex simulation tool can be found in Morcrette et al. (2009), Benedetti et al. (2009), Wagner et al. (2015), Marécal et al. (2015), and Cuevas et. al. (2015).

The MACC/CAMS aerosol parameterization is based on the LOA/LMD-Z (Laboratoire d'Optique Atmosphérique/Laboratoire de Météorologie Dynamique-Zoom) model (Reddy et al., 2005). MACC/CAMS considers five types of tropospheric aerosols: sea salt, dust, organic and black carbon, and sulphate aerosols. Prognostic aerosols of natural origin such as mineral dust and sea salt are described using three size bins. For dust, the size (radius) classes are from 0.03–0.55 $\mu$m, 0.55–0.9 $\mu$m, and from 0.9–20 $\mu$m (see Table 1), whereas for sea salt, the bin limits are at 0.03, 0.5, 5 and 20 $\mu$m (Morcrette et al., 2009). For the production of desert dust in the ECMWF model, a formulation of the source was implemented following the approach of Ginoux et al. (2001). A detailed description of the dust scheme is given in Morcrette et al. (2009).

MODIS (Moderate Resolution Imaging Spectroradiometer) observations of aerosol optical thicknesses (AOT) are assimilated into the model (Benedetti et al., 2009). The AOT used in the MACC/CAMS simulations in 2013 was retrieved by means of a MODIS aerosol analysis scheme which consisted of two entirely independent algorithms, one for deriving aerosols over land (for dark surfaces) and the second for aerosols over ocean (Remer et al., 2005). AOT information for the bright dust source regions were partly not available or highly uncertain. The MACC/CAMS AOT observation operator derives the optical depth based on precomputed aerosol optical properties and model relative humidity for the aerosol species mentioned above. After assimilation, the model output represents the best statistical compromise between the model background (forecast running without assimilation) and observations. The NRT product of MACC/CAMS assimilates MODIS Deep Blue aerosol products (Hsu et al., 2013) since September 2015. Although the total AOT was well constrained in 2013, no attempt was performed to

adjust the specific marine, smoke or dust AOTs. These specific AOTs remained a function of the model characteristics and parameterizations of emission, transport, and deposition.

The MACC reanalysis was stopped in 2012 so that reanalysis products are not available for 2013. Therefore, the operational (forecast) runs for 2013 were used in our study. Data sets of particle mass concentrations were downloaded from the ECMWF data server (MACC, 2016) for a fixed grid with resolution of $1.125°$. We selected the data sets for the nearest grid points to the position of R/V Meteor and 00:00 UTC (cases 2-4) and 04:00 UTC (case 1, 23 May 2013). The model resolution was $0.8° \times 0.8°$ with 60 vertical levels in 2013. Since 2016, the model resolution is increased to $0.5° \times 0.5°$.

As mentioned, dust mass concentrations are available for three radius classes.The radius class from 0.03–0.55 $\mu$m defines the fine dust fraction, the two other size classes belong to the coarse mode. Morcrette et al. (2009) state that the size bins are chosen such that the mass concentration percentages are 10% for the fine dust class, and 20% and 70% for the two coarse dust size bins during emission. For the specific simulations of the May 2013 cases, the released dust fractions were 7.6% (fine dust, 0.03-0.55$\mu$m radius interval), 30.7% (coarse dust, 0.55-0.9 $\mu$m radius interval), and 61.7% for super coarse dust (0.9-20 $\mu$m

radius interval). Thus the mass-related FMF, which considers per definition particles with radius up to 0.5 $\mu$m, was about 0.05-0.07 during emission.

     We calculated the particle extinction coefficients at 550 nm by dividing the MACC/CAMS fine, coarse and total dust mass concentrations by extinction-to-volume conversion factors of $0.21 \times 10^{-12}$ Mm (fine dust), $0.81 \times 10^{-12}$ Mm (coarse dust), $0.64 \times 10^{-12}$ Mm (total dust), and $0.65 \times 10^{-12}$ Mm (marine aerosol) (Mamouri and Ansmann, 2017), respectively, and by

the particle density. MACC/CAMS assumes a particle density of 2.6 g cm$^{-3}$ for mineral dust and of 1.2 g cm$^{-3}$ for marine particles. The used extinction-to-volume conversion factors derived from extended AERONET studies at Morocco, Cabo Verde, and Barbados (Mamouri and Ansmann, 2017) are in full agreement with the defined mass specific extinction coefficients linking the microphysical and the optical properties (at 550 nm) in the MACC/CAMS model given in Benedetti et al. (2009). As in the case of SKIRON (and of NMMB/BSC-Dust, see text below), the MACC/CAMS fine dust class (0.03–0.55 $\mu$m radius)

includes particles with radii exceeding 0.5 $\mu$m so that a weak overestimation of the fine-mode dust extinction coefficients of the order of 10% must be taken into account in the discussions of the results in the next section.

### 3.3  NMMB/BSC-Dust

The NMMB/BSC-Dust model (NMMB: Nonhydrostatic Multiscale Model on the B Grid, BSC: Barcelona Supercomputing Center) is the mineral dust module of the NMMB-MONARCH (MONARCH: Multiscale Online Nonhydrostatic Atmo-

spheRe CHemistry) (Pérez et al., 2011; Haustein et al., 2012; Jorba et al., 2012; Spada et al., 2013; Badia and Jorba, 2014; Basart et al., 2016; Di Tomaso et al., 2017) designed and developed at BSC (Barcelona Supercomputing Center) in collaboration with NOAA NCEP (NOAA: U.S. National Oceanic and Atmospheric Administration, NCEP: National Centers for Environmental Prediction), and the NASA Goddard Institute for Space Studies. The NMMB-MONARCH model considers all relevant atmospheric aerosol types such as dust, sea-salt, sulfates, organic, and black carbon, as well as aerosol-formation relevant gases. The model also has a data assimilation system.

The dust model is online coupled with the non-hydrostatic NMMB model (Janjic et al., 2011, and references therein), which is able to increase the model horizontal resolution up to 1 km. The NMMB/BSC-Dust model provides operational forecast over regional (North Africa, Middle East, Europe) and global domains, and it has been selected by the World Meteorological Organization (WMO) as the operational model for the Barcelona Dust Forecast Center (http://dust.aemet.es/), the first WMO regional center specialized in atmospheric sand and dust forecast. Additionally, the model is participating in the WMO (Sand and Dust Storm Warning Advisory and Assessment System, SDS-WAS, https://sds-was.aemet.es/) and ICAP (International Cooperative for Aerosol Prediction) model intercomparison exercises (Sessions et al., 2015). The MACC/CAMS model also contributes to WMO-SDS WAS and ICAP.

The NMMB/BSC-Dust model assumes a viscous sublayer between the smooth desert surface and the lowest model layer (Janjic, 1994; Nickovic et al., 2001), and its dust emission caused by surface and turbulence winds are physically-based on an emission scheme which explicitly considers saltation and sandblasting processes (White, 1979; Marticorena and Bergametti, 1995; Marticorena et al., 1997). The model uses soil texture data from the hybrid STATSGO-FAO (STATGO: State soil geographic data base, FAO: Food and Agriculture Organization of the United Nations) soil map and land use data of the USGS (United States Geological Survey). According to the criteria used in Tegen et al. (2002), the model considers 4 soil populations (i.e. clay, silt, fine-medium sand and coarse sand). The dust vertical flux is distributed following D'Almeida et al. (1987) and then distributed over each of the 8 dust size transport bins (0.1-10 $\mu$m, see Table 1). Mineral dust source areas are defined through the topographic preferential source approach (Ginoux et al., 2001) and the National Environmental Satellite, Data, and Information Service (NESDIS) vegetation fraction climatology (Ignatov and Gutman, 1998). A comparison of the main features of NMMB/BSC-Dust and the dust scheme of MACC/CAMS can be found in Huneeus et al. (2016).

The NMMB/BSC-dust model solves the mass balance equation and includes parameterizations for the gravitational settling and dry deposition at the first layer (Zhang et al., 2001), and wet deposition by sub-cloud and in-cloud scavenging from convective and stratiform clouds (Betts, 1986; Betts and Miller, 1986; Janjic, 1994; Ferrier et al., 2002). The model has been evaluated at regional and global scales (Pérez et al., 2011; Haustein et al., 2012; Gama et al., 2015; Binietoglou et al., 2015; Huneeus et al., 2016; Basart et al., 2016) showing the ability of the model to reproduce the dust cycle.

For the current intercomparison exercise, a global experiment with $1.4° \times 1°$ horizontal resolution and 40 hybrid layers is considered. The simulated dust distributions consist of daily (24 hours of forecast) runs for 25 April to 23 May 2013. The NCEP/FNL final analyses (at $1° \times 1°$) at 0 UTC are used as initial meteorological conditions. The model did not include a dust data assimilation system. The initial state of the dust concentration is defined by the 24-h forecast of the previous-day model run. Only in the 'cold start' of the model (here on 28 April 2013), the dust concentration is set to zero. In this contribution, simulations were carried out with the operational RRTM (Rapid Radiative Transfer Model) radiation scheme (Mlawer et al., 1997) which allows feedback between dust and radiation. Simulated fields of dust mass (per bin of the model) with a temporal resolution of three hours are used to obtain the bilinear temporal and spatial interpolated simulated profiles of the NMMB/BSC-Dust model following the transatlantic path (Fig. 1). Dust extinction coefficients at 550 nm are computed from the simulated dust mass concentration profiles in the same way as described in Sect. 3.1 for the SKIRON simulations.

## 4 Comparisons

### 4.1 Marine and dust aerosol profiles: MACC/CAMS simulations vs lidar observations

In Fig. 5, MACC/CAMS simulations of marine and dust profiles are compared with respective lidar observations for the four selected key scenarios (cases 1-4 in Fig. 1). We removed the smoke and haze contribution from the lidar-derived non-dust extinction profile in the SAL (above 2 km height) in case 2. The non-dust extinction contribution in the SAL in cases 3 and 4 may be caused by aged smoke, but we cannot exclude that this aerosol is of marine origin after long-range transport over the Atlantic. As described in Rittmeister et al. (2017), the lofted dust in the SAL observed on 5 May 2013 (case 4) traveled about 9 days across the Atlantic (4300 km) before reaching the research vessel at 53°W at relatively low heights. The dust air masses observed on 9-10 May (case 3) needed 5 days for the 3300 km travel from the west coast of Africa to the R/V Meteor. The dust layer on 14-15 May (case 2) and 23 May (case 1) needed 3 days and one day across the Atlantic (1700 km), respectively, before reaching the lidar site. The strong dust layers observed on 23 May at Cabo Verde (case 1) were advected directly from desert areas north of 15°N so that the impact of anthropogenic haze and biomass burning smoke was low.

As can be seen in Fig. 5, the dust load and vertical distribution is well simulated in cases 1 and 2. There is very good agreement regarding both, the marine aerosol profile and the dust profiles in case 2. The model results deviate considerably from the lidar observations in cases 3 and 4, disregarding the uncertainty regarding the non-dust (marine or smoke/haze) extinction contribution in the SAL. Before discussing the differences between the observations and the model results in more detail, it should be emphasized that the dust layering observed with lidar remained unchanged over hours (Rittmeister et al., 2017) in all four cases so that we can conclude that the lidar observations are representative for the dust conditions of a larger area around the R/V Meteor (100 km×100 km) and thus appropriate for comparison with the modeling results.

We explicitly checked the variability in the MACC/CAMS dust mass concentration profiles and downloaded 9 profiles around the R/V Meteor for cases 1–4. Besides the profile of the mass concentration profile closest to the ship, shown in Fig. 5, we checked the other 8 different model profiles at gridpoints surrounding the ship location. As a result, the modeled dust profiles varied within 20–30% (standard deviation, case 1), 10–20% (case 2), 50-80% (case 3), and 20-30% (case 4) for the height range of the SAL.

The lidar-derived overall (marine + dust) AOT values for cases 1–4 are 0.63, 0.26, 0.11, and 0.11, respectively, and close to the ones obtained by integrating the MACC/CAMS extinction profiles of marine aerosols and dust particles over the vertical column and adding both column values. MACC/CAMS AOTs are 0.54 in case 1, 0.28 in case 2, 0.12 in case 3, and 0.15 in case 4. As mentioned above, MACC/CAMS was assimilating MODIS AOT data in 2013. However, no attempt was undertaken to adjust the marine and dust contributions to the total AOT, for example by considering typical marine extinction profiles as a constraint. As shown in Fig. 5, the strong underestimation of the dust load in cases 3 and 4 is compensated by unrealistic profiles for the marine aerosol to match the MODIS AOT. Case 4 shows a strong marine layer up to 3 km and a marine AOT of 0.16, whereas the reality is reflected in the lidar observations with a top height of the marine boundary layer (MBL) at 1 km height and an overall (MBL and free tropospheric) marine AOT of 0.06. The lidar indicates marine particles mainly below 1 km height and marine AOT of 0.09, 0.1, 0.05, and 0.06 for cases 1–4, respectively. MACC/CAMS simulates marine AOTs

of 0.15-0.16 for case 1,2 and 4, and about 0.1 for case 3. Thus, the quality of the dust forecast clearly decreases with distance from Africa and increasing dust travel time. However, MACC/CAMS predicts the SAL height range (from bottom to top) very

well in cases 1–3.

Efforts are undertaken to improved global dust cycle modeling by assimilation of higher-level spaceborne AOT products into the forecast models (see, e.g., Huneeus et al., 2012; Di Tomaso et al., 2017; Escribano et al., 2017). The consideration of MODIS (Deep Blue) AOT products is promising. It is expected that the dust forecasts, especially over continents (including the bright dust source regions) will be significantly improved.

Concerning the uncertainties in the lidar profiles in Fig. 5, we should add for completeness that the dust lidar retrieval may be slightly affected by enhanced light depolarization, when wet, spherical sea salt particles get dried and then become almost cubic in shape (Haarig et al., 2017b). This occurs usually at the top of the MBL where the relative humidity drops from values around 80% to <50% in the free troposphere. The retrieved dust mass concentration is then too high when assuming spherical marine particles and a marine depolarization ratio of 0.02–0.03. Cubic sea salt particles can cause particle depolarization ratios

of 0.1–0.15. The dry marine particle effect may be responsible for the small peaks in the dust extinction profiles at MBL top coinciding with the SAL bottom in cases 1 and 2. The effect is however in the 5-10% range regarding the underestimation of the marine AOT and thus can be neglected in our model-observation comparisons.

### 4.2   Dust mass profiles: MACC/CAMS, NMMB/BSC-Dust, and SKIRON simulations vs lidar observations

In Fig. 6, we compare the dust mass concentrations obtained with the three models with the respective lidar observations. As

a main result, good agreement with the lidar observations were obtained in case 2 (MACC/CAMS, NMMB/BSC-Dust). In case 1, both global models underestimated the total atmospheric dust burden, most probably linked to a too low dust emission in the model. SAL top and base heights were well predicted in cases 2 and 3. However, the two global models considerably underestimate the dust load in cases 3 and 4. The vertically-integrated dust mass concentrations (numbers are given in Fig. 6) are a factor of 2 (case 3, 3300 km west of Africa) and 3 (case 4, 4300 km west of Africa) smaller than the measured ones

in the case of the MACC/CAMS and NMMB/BSC-Dust simulations. We analyzed METEOSAT satellite observations for the potential impact of wet deposition. As mentioned in part 1 (Rittmeister et al., 2017), we analyzed METEOSAT satellite observations for the presence of strong cumulus convection and found that, except for case 4, wet deposition by deep convection and associated rain can be excluded. However, fair weather cumulus convection and light precipitation always occurs over the tropical Atlantic and thus a certain contribution of wet deposition to dust removal must be always taken into account.

In contrast to the column mass values obtained with the global models, the ones simulated with the regional model SKIRON are in good agreement with the lidar values. However, a systematic shift of the dust maximum mass concentration towards lower heights (compared to the lidar profiles) is observed in the SKIRON profiles in all of the four observational cases. As discussed in Sect. 4.3, the ratio of fine dust to coarse dust mass fraction is set constant throughout the simulation period in the SKIRON model and the fine-to-coarse dust fraction is generally lower in SKIRON simulations than in the forecasts obtained with the global models (NMMB/BSC-Dust, MACC/CAMS). As a result of the higher coarse dust fraction, gravitational settling obviously has a higher impact on the dust profile simulated with the regional SKIRON model.

It is impossible to identify the reason for the too strong removal of dust in the global models. Too many sources for uncertainties exist. Many experimental and empirically derived constants are used in the source/emission, transport, and deposition parameterizations so that simple conclusions concerning the reasons for the found uncertainties can not be drawn. The differences in forecasts can be large. Huneeus et al. (2016) compared simulations of five different models in the case of a strong Saharan dust outbreak towards northern Europe and found differences between the largest and the smallest dust emissions of the order of a factor of 10. As Kok et al. (2014a, b) mentioned, the simulations of the global atmospheric dust cycle are hindered by the empirical nature of the presently widely used dust emission (or source) parameterizations in weather and climate models. These parameterizations are generally tuned to reproduce the current dust cycle (for given present climate conditions, global circulation pattern, land use, and surface characteristics), and thus can introduce large uncertainties for specific, individual, regional meteorological conditions and individual dust outbreak scenarios.

A too strong dust removal effect was observed in many studies (e.g., Kim et. al., 2014; Mona et al., 2014; Binietoglou et al., 2015). Kim et. al. (2014) found a systematic underestimation of the dust load by a factor of 2 from the African coast downwind to Barbados according to the satellite observations (MODIS and MISR: Multi-angle Imaging SpectroRadiometer), whereas the five involved models produced a decay by a factor of roughly 4–10 from the west coast of Africa to 60°W. They also found large intermodel diversities. The retention of the coarse mode particles was also noticed in recent airborne Saharan dust studies by Ryder et al. (2013) and Weinzierl et al. (2017).

One possible reason for the too strong dust removal in the models was discussed in part 1 (Rittmeister et al., 2017). The R/V Meteor lidar observations, the SALTRACE observations at Barbados, as well as spaceborne lidar measurements (CALIOP: Cloud Aerosol Lidar with Orthogonal Polarization) over the tropical Atlantic suggest that other processes besides gravitational settling of dust must be active in the SAL, and weaken the dust removal strength caused by fall out of dust particles (Ulanowski et al., 2007; Nicoll et al., 2011; Yang et al., 2013; Colarco et. al., 2003; Gasteiger et al., 2017; Haarig et al., 2017a). During SALTRACE, the lidars generally observed an almost height-independent vertical profile of the dust-related particle depolarization ratio within the SAL which indicates well-mixed conditions rather than an accumulation of larger particles in the base region of the SAL as would be the case as a result of gravitational settling. A higher amount of coarse particles in the lower part of the SAL would lead to a systematic decrease of the particle depolarization ratio from SAL base to top. Gasteiger et al. (2017) argue that absorption of solar radiation introduce turbulent mixing of dust within the SAL which weakens the pure sedimentation-based removal effect. Yang et al. (2013) discusses the possibility that different shapes of small (less irregularly shaped) and large particles (more irregularly shaped) may have an impact on falling speed and thus the vertical dust distribution. Ulanowski et al. (2007) observed that dust layers have an impact on the atmospheric electric field, and argue that dust particles can become charged (when colliding with themselves or the underlying surface), and may be vertically aligned in the electric field, and conclude that these charging effect influence the downward transport of dust.

In the companion paper of Rittmeister et al. (2017), it is further mentioned that the observed particle extinction coefficient in the SAL was always 50–100 Mm$^{-1}$ (cases 2–4) and the particle depolarization ratio showed vertically homogeneous dust conditions. In addition, the wavelength dependence of particle extinction and backscattering, which remained unchanged within the SAL during the long-range tranport, does not indicate a significant change in the dust size characteristics in the SAL during

the travel across the Atlantic. If gravitational settling would dominate, we should see a clear decrease of the dust extinction

coefficient with time, a decrease of the depolarization ratio from SAL base to SAL top as discussed in part 1 (Rittmeister et al., 2017), and also an increase of the wavelength dependence of the optical properties in the SAL.

### 4.3   Fine-mode and coarse-mode dust profiles: simulation vs observation

We compare, to our knowledge for the first time, fine dust and coarse dust profiles derived from the lidar observations with simulated ones. Huneeus et al. (2011) emphasized the need for height-resolved observations of dust-size-characterizing pa-

rameters. In their overview and review paper the authors stated that the dust extinction coefficient and the corresponding dust AOT-related radiative effects are sensitively controlled by the amount of occurring fine dust particles due to their higher extinction efficiency, whereas coarse dust dominates the surface concentration, deposition, and removal. In a recent study, Kok et al. (2017) emphasize in detail the consequences of a not-well modeled dust size distribution and abundance for the global energy balance through direct interaction of dust with radiation. As mentioned, the indirect climate effect of dust through interaction

with clouds is also affected if the dust size distribution and load and thus estimates of CCN and INP concentrations are wrong in the forecast models.

Kok (2011a, b) stimulated a discussion on the dust mobilization parameterization which may have strong consequences for the fine-mode and coarse-mode fractions of the atmospheric dust burden during the entire life cycle. Dust emission according to the theory of saltation and sandblasting predict that the size of emitted dust aerosols decreases with wind speed (Shao,

2001, 2004; Alfaro and Gomes, 2001), whereas the brittle fragmentation theory of dust emission predicts that the emitted dust particle size distribution is independent of the wind speed (Kok, 2011b). Dust emission following the saltation/sandblasting parameterization leads to a size distribution with a comparably strong fine dust fraction and a less pronounced coarse dust fraction. In contrast, according to the brittle fragmentation theory larger coarse dust (particles with diameters exceeding 5 $\mu$m) dominates the emitted size spectrum (Kok, 2011a, b; Kok et al., 2017). As describe in Mahowald et al. (2014), the size distri-

bution following brittle fragmentation theory from Kok (2011a) prescribes mass percents of 1.1, 8.7, 27.7, and 62.5% at every grid point which acts as a source for the 4 bins or size classes with particle diameters from 0.1–1 $\mu$m, 1–2.5 $\mu$m, 2.5-5 $\mu$m, and 5-10 $\mu$m, respectively. This mass spectrum was found to be in good agreement with observations of size-resolved dust mass concentrations during emission (Kok, 2011a). Thus the mass-related fine mode fraction (FMF) may be close to 0.01 rather than 0.07–0.1 as assumed in the SKIRON simulations and 0.05–0.07 in the MACC/CAMS and NMMB/BSC-Dust models. How-

ever, dust emission size distribution may vary strongly as a function of soil characteristics and meteorological conditions so that it remains open in the following discussion to what extent uncertainties in the emitted dust size distribution are responsible for the found differences between observed and modeled fine-to-coarse dust mass ratios after long-range transport illuminated below. A discussion on strong variations and changes in the Saharan dust size distribution observed with aircraft close to the Saharan dust emission zones in fresh and moderately aged dust layers and aged dust layers over the Canary Islands is given by Ryder et al. (2013).

Figures 7 and 8 now show the comparisons between the modeled and observed fine dust and coarse dust extinction and mass concentration profiles. As we mentioned above, our lidar observations are in full agreement with the 500 nm AOT fine dust

fractions observed with sun photometer during the cruise as shown in Fig. 9, and thus reflect very well the true fine-to-coarse dust extinction and mass conditions. The aircraft observation of the dust size distribution during the SALTRACE transfer flights from Africa to the Caribbean on 17-22 June 2013 also reveal fine mode fractions (for 532 nm extinction) of 0.25 in the Cabo Verde region and 0.2 over Barbados (Weinzierl et al., 2017).

In terms of mass concentrations the fine dust profile values are much smaller in Fig 7 than the coarse dust values. In contrast the fine dust and coarse dust profiles are close together in the case of light extinction. Correspondingly the extinction-related FMF is much larger than the mass-related FMF. The mass-related and extinction-related FMF are linked by the so-called extinction-to-volume conversion factors (Ansmann et al., 2012; Mamouri and Ansmann, 2017). For a typical dust size distributions after long range transport, the extinction-to-volume conversion factor is $0.21 \times 10^{-12}$ Mm for fine dust at 532 nm and $0.81 \times 10^{-12}$ Mm for coarse dust. Expressed in optical efficiency, the fine dust particles are a factor of 4 more efficient than the coarse dust particles. Assuming the same dust particle density of fine and coarse dust, the extinction-related FMF of dust is thus about a factor of 2–3 higher than the respective mass-related FMF for dust after long-range transport and dust FMF in the range of 0.1–0.5.

As can be seen in Fig. 7, the fine dust and coarse dust contributions to light extinction are almost equal (1:1) in the NMMB/BSC-Dust simulations of cases 2–4, whereas the lidar observations reveal a much lower value of fine-to-coarse dust light extinction ratio (about 1:4). The simulated contributions of fine dust to total dust light extinction coefficient are strongly overestimated. It should be emphasized again that the simulated size distribution (eight size classes) in the NMMB/BSC-Dust model consist of a pronounced coarse mode and the fine dust contribution originate from the particles in the small-particle wing (size classes 1-3).

The results are similar in the case of the MACC/CAMS simulations in Fig. 8 (top). The fine dust contribution to light extinction is overestimated when compared with the lidar observations and the ratio of fine-to-coarse dust extinction coefficient steadily increases with transport distance from Africa. In contrast, SKIRON (Fig. 8 (bottom)) is in comparably good agreement with the lidar observations. However, this result is obtained by setting the fine-to-coarse dust mass ratio to a constant value throughout the simulations.

Figure 9 provides an overview of the extinction-related FMF for the 500-550 nm wavelength range. Lidar and AERONET values for the four cases 1–4 are compared with SAL mean values obtained from the MACC/CAMS and NMMB/BSC-Dust forecast profiles. The lidar-derived dust FMF value of 0.2 follows from the analysis of the polarization lidar observations (Mamouri and Ansmann, 2017). This value is consistent with the column observations (AERONET) of FMF close to Africa (cases 1 and 2) where dust dominates. Over the remote Atlantic (cases 3 and 4), the AERONET values increase towards 0.3–0.35 because of the increasing impact of marine particles. For pure marine conditions, we found column FMF values close to 0.4 during the R/V Meteor cruise. The comparison of the observed and modeled dust FMF once again clearly reveal a systematic overestimation of the fine dust fraction by the models, and the deviation of the modeled from the observed dust FMF value increases from case 1 to case 4. Since the overestimation is already strong over Cabo Verde (close to the dust source regions), it seems to be obvious that this overestimation is related to the overestimation of the emitted fine dust fraction in the models.

However, as pointed out in the foregoing section, we should keep the discussion open. The emitted dust size spectrum may be one reason. There are many other uncertainty sources in the models. It is impossible to identify in a straight forward way the reason for the differences between the observed and modeled fine-to-coarse dust fraction. Note in this context that models in general seem to have an enhanced numerical diffusion in the sedimentation schemes and for that reason they remove the aerosol burden from the atmosphere faster than it is the case in reality, and coarser particles are much faster removed than the finer ones. Numerical diffusion has a bigger effect on sedimentation than, for example, the effect of particle shape (spherical vs irregularly shaped particles) (Ginoux, 2003).

## 5 Conclusions

A unique observational data set of Saharan dust profiles, measured with ship-based state-of-the-art polarization/Raman lidar over the tropical Atlantic, was compared with dust forecasts of a regional (SKIRON) and two global atmospheric models (MACC/CAMS, NMMB/BSC-Dust). Undisturbed dust transport and removal conditions could be studied in detail. As new feature, the recently introduced lidar data analysis scheme POLIPHON allowed us to retrieve height profiles of dust light-extinction coefficient and mass concentration separately for fine dust, coarse dust, and the non-dust (marine, haze, smoke) aerosol components, and thus to study and discuss the uncertainties in the modeled dust size distribution characteristics during long-range transport. In part 1 (Rittmeister et al., 2017), we presented a dense set of height-resolved observations of dust optical properties and layering structures and found good agreement of the observations with the basic features of the conceptual model of Karyampudi et al. (1999) which describes the long-range transport of dust from western Africa to North America (based on 50 years of dust and SAL research) so that we can conclude that our shipborne lidar observations were representative for typical dust scenarios over the tropical Atlantic.

From the comparisons of the lidar observations with the forecasts of the dust models we can draw three main conclusions: (1) We found good to reasonable agreement between simulations and observed dust profiles (total dust mass concentration and 500-550 nm extinction coefficient) within the transport range of about 2000 km downwind of Africa and thus about 2500-4000 km west of the Saharan dust source regions. The quality of the simulation results decreased significantly with further distance from the source regions. (2) The removal of dust from the atmosphere is too efficient in the models. The main process of vertical exchange (particle downward motion) in the comparably well stratified SAL is assumed to be particle settling via sedimentation. However, as already discussed in detail in part 1, there must be further mechanisms that retain the mostly coarse dust particles in the atmosphere during long-range transport. (3) The highlight of the study was the comparison of the observed (estimated) ratio of fine-to-coarse dust mass concentration and extinction coefficient with the model forecasts. It was found that the models considerably overestimate the fine dust fraction. This aspect is not new (Kok, 2011a; Mahowald et al., 2014). One of the reasons is obviously that the emitted fine dust fraction is already significantly overestimated in the models. However, the complex model structures (including the differences regarding the meteorological drivers in the three models) and the used large set of empirical constants in the emission, transport, and deposition parameterizations made it however impossible to unambiguously identify the reasons for the observed partly systematic biases in the model forecasts. Many points concerning

model performance and uncertainties caused by the implemented aerosol and dust parameterization schemes thus remain open for further investigations.

As an outlook, we recommend to design more atmospheric field campaigns on dust deposition features. More observations are needed to study long-range transport and dust removal and to clarify the role of different mechanisms which are potentially able to prolong the atmospheric lifetime of coarse dust particles. As already mentioned in part 1 (Rittmeister et al., 2017), one of the activities could be long-range airborne in situ dust measurements over thousands of kilometers across the Atlantic with profiling from SAL base to top every 500–1000 km. The measured quantities should be the dust size distribution, mineralogical (chemical) composition, mixing state (internal or external mixtures of dust, smoke, and marine particles), and properties indicating particle shape and composition changes by chemical aging and cloud processes. A first attempt was recently presented by Weinzierl et al. (2017). Other areas of interest for such field studies would be the Eastern Mediterranean, the Middle East, central, and eastern Asia. More airborne and lidar observations close to dust sources such as presented by Ryder et al. (2013) and in the near range of the long-range transport regime (case 1 in our discussion) are also needed to improve our knowledge about dust emissions and the size spectrum during dust release.

Finally, it would be desirable to continuously work on the establishment of a comprehensive dust assimilation scheme which ideally would include regular observations from space (passive and active remote sensing), with AERONET, observations with the worldwide ceilometer network (organized by the weather services), existing ground-based networks of cheap, robust, continuously operating standard backscatter lidars, as well as monitoring with continuously running advanced aerosol/dust lidars (e.g., of the Polly type) at distinct stations in key regions of dust occurrence and transport.

## 6 Data availability

The R/V Meteor lidar data are available at TROPOS upon request (info@tropos.de). The NMMB/BSC-Dust dust profiles are available upon request (http://www.bsc.es/ESS/bsc-dust-daily-forecast, e-mail: info-services-es@bsc.es). The MACC/CAMS dust profiles are downloaded from the MACC aerosol data base (MACC, 2016). The SKIRON dust profiles can also be provided upon request and are also available at *http://forecast.uoa.gr/dustindx.php*.

*Acknowledgements.* We thank the R/V Meteor team and German Weather Service (DWD) for their support during the cruise M96. We appreciate the effort of AERONET MAN to equip research vessels with sun photometers for atmospheric research. We are grateful to Angela Benedetti (European Center for Medium Range Forecast, Reading UK, MACC/CAMS model) for all her fruitful comments and suggestion during the initial phase of paper preparation. The SKIRON model operations were supported by the European Commission through the 7th Framework Program MARINA Platform (Marine Renewable Integrated Application Platform, Grant Agreement 241402). NMMB/BSC-Dust model simulations were performed in the MareNostrum supercomputer hosted by BSC. S. Basart and O. Jorba acknowledge the CICYT project (CGL2016-75725-R) of the Spanish Government and the AXA Research Fund.

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

825

**Table 1.** Size bins (radius intervals) considered in the dust forecast models MACC/CAMS, SKIRON and NMMB/BSC-Dust

| Model | fine dust size bins | coarse dust size bins |
|---|---|---|
| MACC/CAMS | 0.03–0.55 $\mu$m | 0.55–0.9 $\mu$m |
| | | 0.9–20 $\mu$m |
| NMMB/BSC-Dust, SKIRON | 0.1–0.18 $\mu$m | 0.6–1.0 $\mu$m |
| | 0.18–0.3 $\mu$m | 1.0–1.8 $\mu$m |
| | 0.3–0.6 $\mu$m | 1.8–3.0 $\mu$m |
| | | 3.0–6.0 $\mu$m |
| | | 6.0–10.0 $\mu$m |

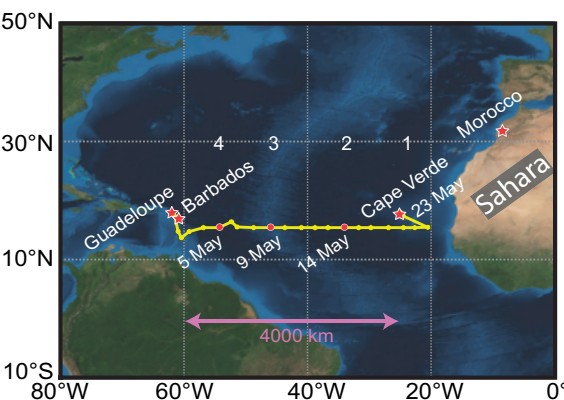

**Figure 1.** The cruise track of the R/V Meteor from Guadeloupe (29 April 2013) to Cabo Verde (23 May 2013) indicated as a yellow line (Rittmeister et al., 2017). Lidar observations 1–4 (see red circles and star at Cabo Verde) representing key stages of dust layering over the remote Atlantic are discussed in detail in part one (Rittmeister et al., 2017) and are compared with simulated dust profiles in Sect. 3. The locations of the lidar observations are 1000 km (case 1, 23 May 2013, 03:45–05:00 UTC), 1700 km (case 2, 14 May 2013, 23:45–00:20 UTC), 3300 km (case 3, 9 May 2013, 23:15–24:00 UTC), and 4300 km (case 4, 5 May 2013, 23:40–00:35 UTC) west of the African coast.

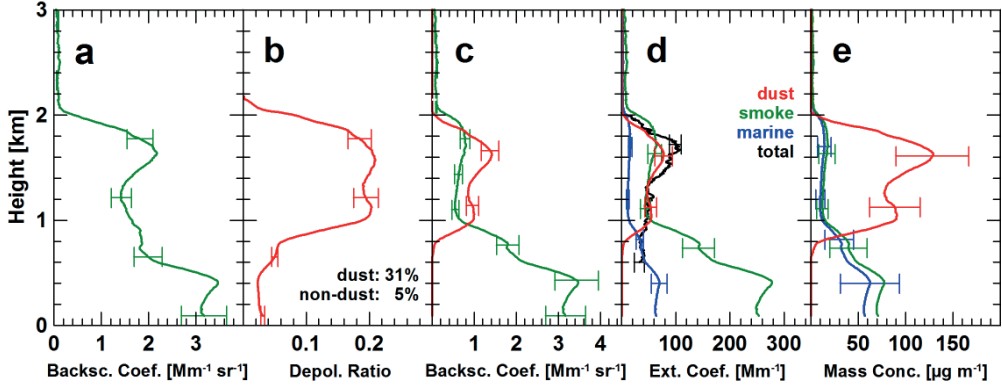

**Figure 2.** Shipborne lidar observations of the Saharan air layer (SAL, 800–2000 m height) above the marine aerosol layer (MAL) on 5–6 May 2013, 23:40–00:35 UTC (see Fig. 1, case 4): (a) 532 nm particle backscatter coefficient, (b) 532 nm particle linear depolarization ratio, (c) non–dust (green) and dust (red) particle backscatter coefficients, (d) dust extinction coefficient (red) and non-dust extinction coefficients (blue, if the non-dust component is of marine origin, lidar ratio of 20 sr; green, if the aerosol component is, e.g., biomass burning smoke and anthropogenic haze, lidar ratio of 50 sr), and (e) dust (red) and non–dust mass concentration (blue, if marine, green if smoke and haze). Error bars indicate the retrieval uncertainties (one standard deviation). The black curve in (d) it the total particle extinction coefficient independently determined by means of the Raman lidar method.

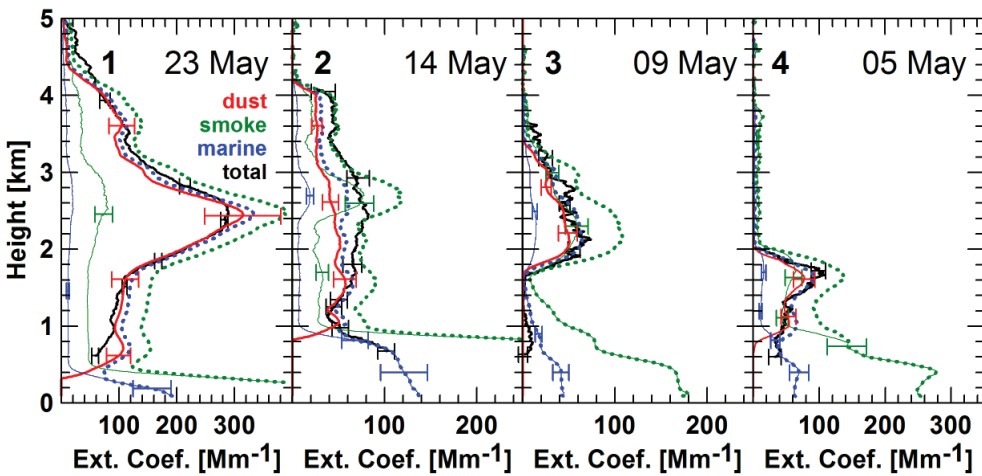

**Figure 3.** The Saharan aerosol layer as a function of distance from Africa (see Fig. 1, cases 1–4). Shown are the 532 nm particle extinction coefficient measured with Raman lidar (black curves) on 5–6 May, 23:40–00:35 UTC (case 4), 9 May, 23:15–24:00 UTC (case 3), 14–15 May, 23:45–00:20 UTC (case 2), and (d) 23 May 2013, 03:45–05:00 UTC (case 1), and the extinction contributions by desert dust (red solid line) and marine (blue solid line, lidar ratio of 20 sr) or smoke and haze particles (green solid line, lidar ratio of 50 sr). The sum of dust and non-dust extinction contributions are given as thick dotted lines. If the black extinction curve is close to the blue dotted line the non-dust aerosol component is probably of maritime origin (case 1,3,4), in the case that the black and green dotted lines match, the non-dust aerosol component is most likely of anthropogenic origin (upper part in case 2).

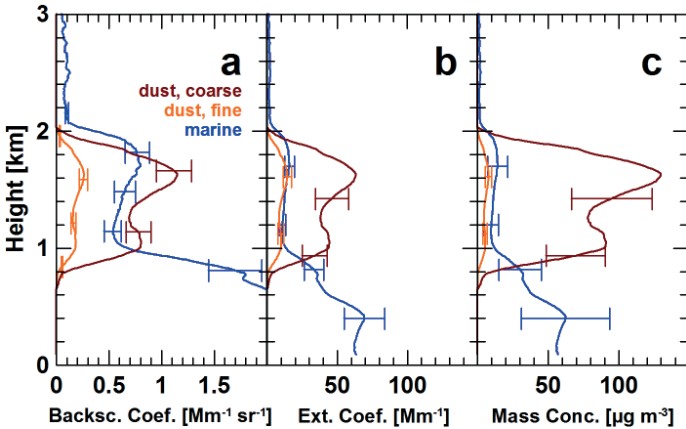

**Figure 4.** Solutions of the 2-step POLIPHON data analysis (for case 4):(a) non–dust (blue, here for marine aerosol with lidar ratio of 20 sr), fine-mode dust (orange) and coarse-mode dust (dark red) backscatter coefficients, (b) respective marine, fine-mode, and coarse-mode dust extinction coefficients, and (c) marine, fine-mode and coarse-mode dust particle mass concentration. Error bars indicate the retrieval uncertainties (one standard deviation).

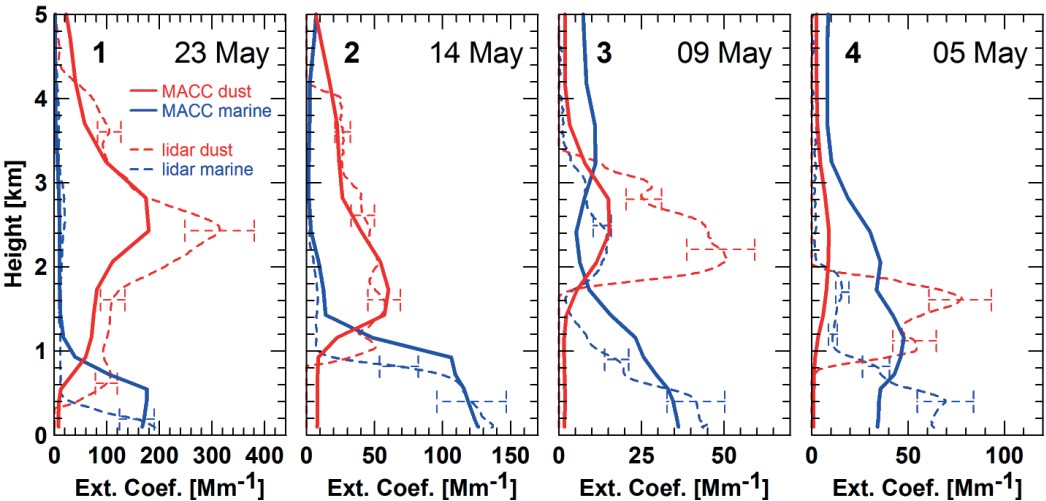

**Figure 5.** MACC/CAMS simulated marine (blue solid) and dust (red solid) contributions to particle extinction coefficient versus respective lidar observations (dashed blue and red lines) for cases 1–4. Error bars show the overall retrieval uncertainty in the lidar observations. The variability in the modeled dust profiles around R/V Meteor are estimated to be 20-30% (case 1), 10-20% (case 2), 50-80% (case 3), and 20-30% (case 4, see text for more details).

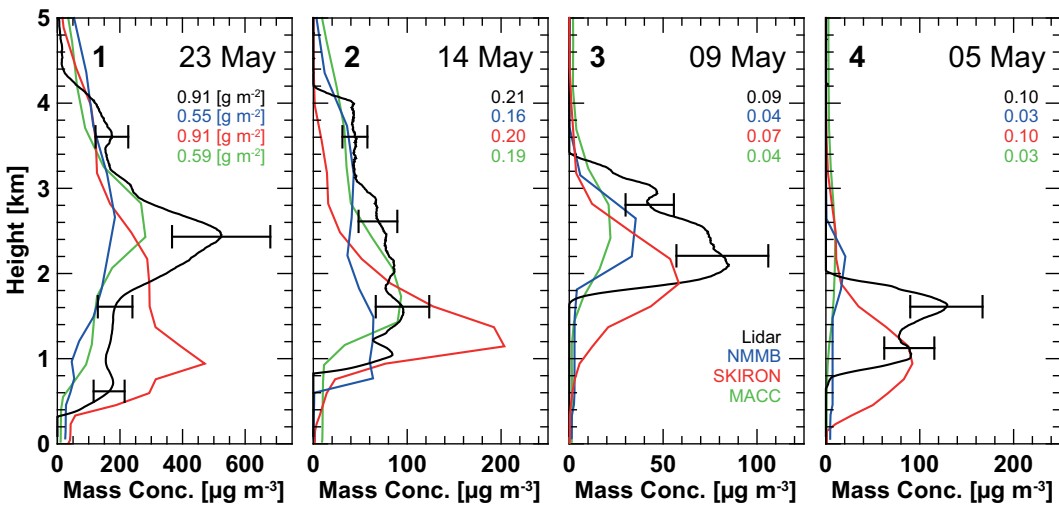

**Figure 6.** Dust mass concentration profiles observed with lidar (black, with retrieval uncertainty bars) and simulated with NMMB/BSC-Dust (blue), SKIRON (red), and MACC/CAMS (green) for cases 1–4. Column-integrated dust mass concentrations are given as numbers.

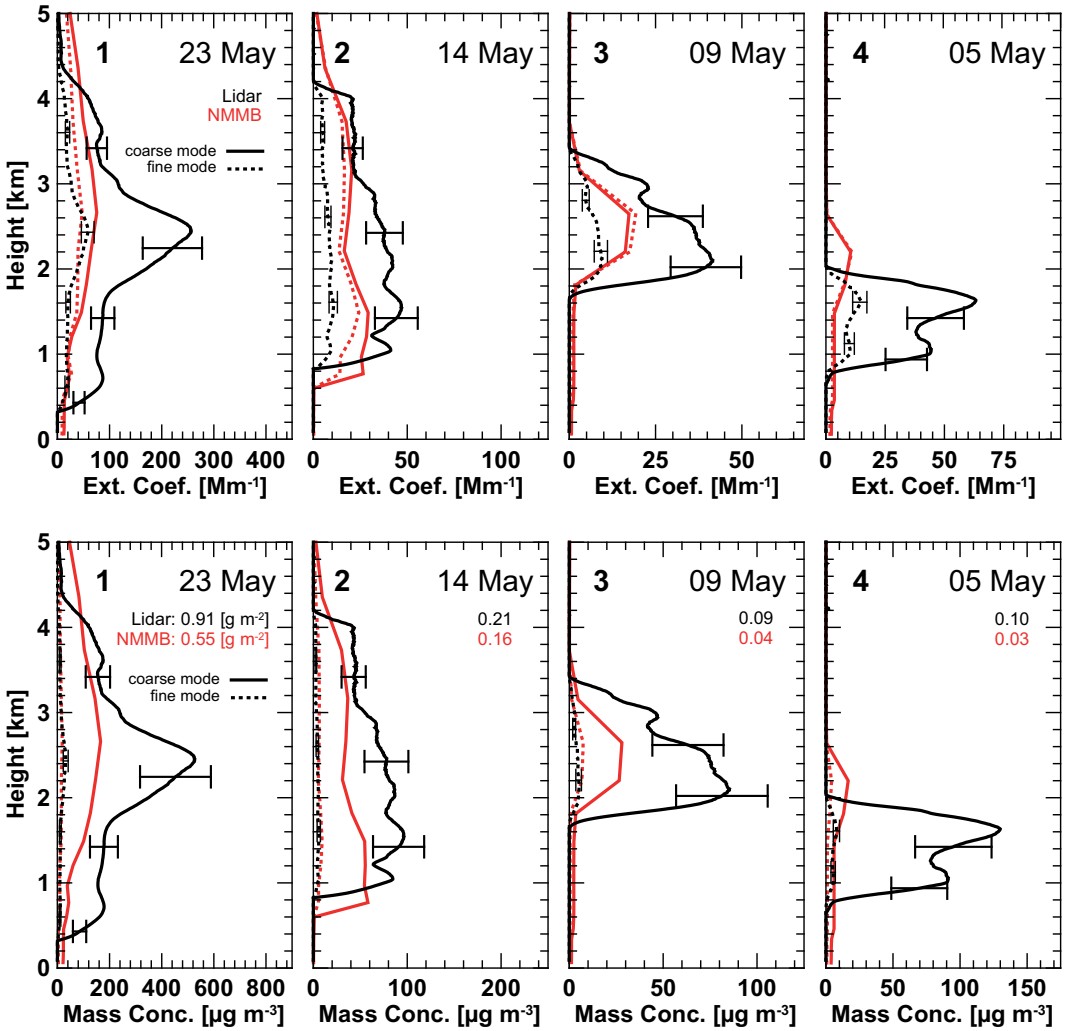

**Figure 7.** (Top) Comparison fine (dotted) and coarse (solid) dust extinction coefficients derived from lidar observations (black) and simulated with NMMB/BSC-Dust (red). (Bottom) Respective fine (dotted) and coarse (solid) dust mass concentrations derived from the lidar measurements (black) and simulated with NMMB/BSC-Dust (red). Column-integrated total dust mass concentrations are given as numbers. NMMB/BSC-Dust profiles are not available for case 1.

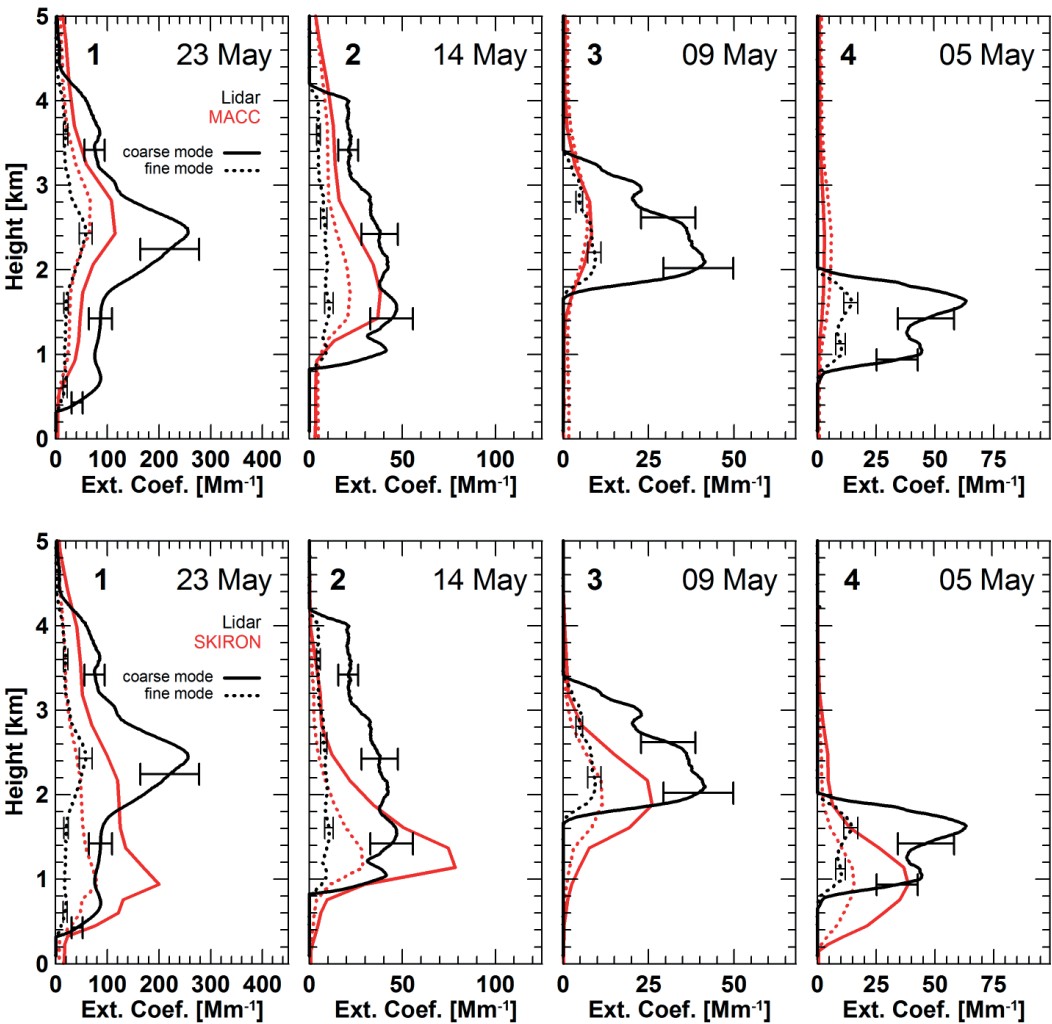

**Figure 8.** Same as Fig. 7 top, except for comparison with MACC/CAMS simulations (top) and SKIRON (bottom) for all four cases.

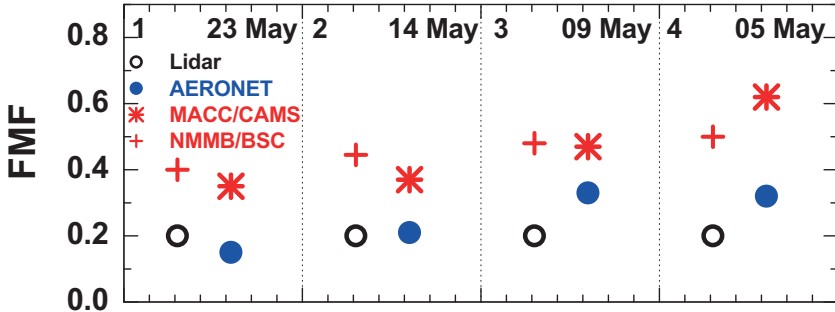

**Figure 9.** Comparison of modeled (MACC/CAMS, NMMB/BSC-dust) and lidar-derived SAL dust fine mode fraction (extinction-related FMF). AERONET sun photometer observations for the entire vertical column (MAL + SAL) and thus influenced by marine, dust and smoke particles are shown in addition.