# Peer review of "Profiling of Saharan dust from the Caribbean to western Africa – Part 2: Shipborne lidar measurements versus forecasts"

_Atmospheric Chemistry and Physics, 2017_

## Referee Comment (RC1) · Anonymous Referee #1 · 2 Aug 2017

This is an interesting paper that reports comparisons of simulated and (lidar) measured particle extinction profiles across the tropical atlantic. It's an excellent addition to the literature, as such lidar measurements have not previously been available to dust modelers. The results presented here provide an independent line of evidence for two conclusions that the dust community has been converging towards: (1) dust is too fine in most models, and (2) dust (and especially coarse dust) in the Saharan Air Layer settles out of models too quickly. I recommend that this article be published after some moderate revisions.

Comments:

- The writing can be improved in places. In particular, at the beginning of sections 2 and 3 a brief explanation of what is to follow in that section and how that builds towards addressing the main objective would be helpful. Related to that, reiterating the paper's main objectives in those key locations would help focus the reader.

- I cannot comment on the validity of the lidar retrievals or the POLIPHON method, as this is outside of my expertise. This is quite critical to the paper, so I hope another reviewer on this paper or the companion paper can comment on this.

- The constants that convert measured backscatter coefficients to light extinction coefficients are given as single numbers. However, the light extinction results reported in this paper are quite sensitive to these conversion constants, and so it's important that the authors provide realistic uncertainties on these numbers (and propagate them) so that their results can be interpreted correspondingly.

- P. 5, l. 18-22: The description here of how the extinction-to-volume conversion factors are obtained is a bit problematic, because the cited paper (Mamouri and Ansmann, 2017) is in prep and not accessible. So more explanation should be given here. Does this conversion account for dust asphericity or does it assume spherical particles? Also, references should be included for the assumed particle densities.

- P. 5, l. 25-26: The reader is referred here again to an in prep manuscript (Mamouri and Ansmann, 2017) for details on the methodology to separate fine and coarse dust, so I'd suggest replacing this reference with the published Mamouri and Ansmann (2014). Since this is a key component of the methodology, a more detailed review of this method would be very helpful to the reader. It's unclear to me how the different depolarization ratios allow a decomposition of the signal into fine dust, coarse dust, and non-dust. That still seems like an underconstraint problem, so more explanation is necessary here.

- P. 5, l.28-30: Assuming a height-independent size distribution is an important simplification that requires experimental support. Is this shown by other SALTRACE measurements? Some references are needed here.

- P. 5, l.30-31: the authors here claim "full agreement" with shipborne fine and coarse dust AOT. For this statement to be convincing, it should be shown, especially as the authors use this to further assert that their lidar observations "reflect very well the true fine-to-coarse dust extinction and mass conditions." Otherwise, these statements should be removed.

- Section 3: Since the simulated fine and coarse dust abundances are compared against measurements in Figs. 7 and 8, the authors should provide the emitted fine and coarse fractions for each model. This helps the reader interpret whether the model discrepancies are due to errors in emission, transport, or deposition.

- P. 13, l. 23-5: I think the authors have enough information to at least hypothesize about the reasons for the model disagreement. For instance, the discrepancy close to source regions, where model errors in transport and deposition are minimal, suggests that models have a problem with their emitted size distribution. The fact that this discrepancy increases with distance from the source region suggests that dust in models is depositing too fast, which the authors already alluded to in other places in the manuscript. I consider these two of the paper's main take-home messages, so I would suggest the authors sum that up here.

---

## Referee Comment (RC2) · Anonymous Referee #2 · 31 Aug 2017

Ansmann et al. Compare lidar retrievals along the E-W transect between West Africa/Cape Verde and Barbados, at fixed distances from the Saharan dust source region, against simulated dust profiles from 3 dust forecast models. What is novel and most interesting is that for the first time the dust fine and coarse mode contributions to the total vertically resolved extinction can be compared in both model and observations. While I cannot comment on the techniques used to decompose the lidar aerosol profiles into the fine, coarse and non-dust profiles (outside of my area of expertise), This potentially offers a new type of dust forecast evaluation methodology and is a potentially very useful observational dataset for the dust forecasting community.

Overall I find this to be an interesting, useful addition to the literature on dust model evaluation and recommend publication following some minor comments below:

Overall the manuscript is well written and clear. There are typos which I haven't listed here (can do so if required) so would recommend a thorough checking through of paper for such errors and mistakes prior to publishing.

P5 L23-29: It would be nice to see the lidar retrievals independently verified if possible. The authors refer to agreement with shipborne sun photometer observations. I would recommend this verification is shown, particularly as the relevant reference Mamouri and Ansmann, 2017 is still in prep.

P6, Section 3.1: Some extra detail is needed in terms of the model description of SKIRON. What region does the SKIRON model cover? What resolution is the model run at – this is important in terms of the skill of the model in simulating the dust emission which is dependent on wind speed. Does the model have its own meteorological data assimilation to constrain the meteorological variables or is it just free-running? Again this is important information in terms of assessing the dust transport. Please include some more information plus a bit more detail on the dust model itself.

Is SKIRON an operational dust forecasting model and therefore dust products were available or was a dedicated experiment conducted, this is not clear from the description here.

P7, L2-3 and L5: The MACC/CAMS simulation tool -> poor choice of wording , would suggest changing tool ->system. The key feature of the MACC/CAMS system is that Earth Observation data are operationally assimilated to provide Near-real time forecasts for a wide range of forecast products - this wasn't clear to me in the opening sentences of Section 3.2 - suggest making it clearer.

P7 L18: My understanding of the MACC system is that they don't yet assimilate Deep Blue retrievals over bright surfaces (ie: not that they were not available as stated in the
text) – I would suggest ensuring your statements here are accurate and again the key point being that MACC do not assimilate AOD over bright desert surfaces.

P7 L20: Again the key point with the MACC aerosol assimilation is that while it is very good at constraining the total AOT, the control variable in the assimilation is the total aerosol mass mixing ratio and the increment to the total mixing ratio needs to be then redistributed across all individual aerosol species which is done based on the model background. So while the total AOD should be well constrained the speciation and therefore dust contribution to the total extinction is not. This is clear on P10 L5-10. I think these key features of the MACC/CAMS system need to be made more clear for the reader.

P7 L24-26: Two different model resolutions are mentioned here which is confusing. Are the authors saying the model simulation is run at 0.8x0.8 deg but output products used in the present analysis were only available at 1.125 deg? Clarification required.

P8 L15-18 The MACC model also contributes to WMO-SDS WAS and ICAP (I'm not sure about SKIRON) so authors should be consistent in descriptions or remove the statements. Appropriate reference for ICAP is Sessions et al: https://www.atmos-chem-phys.net/15/335/2015/acp-15-335-2015.html

P9 L2: Was the model initiated from 0 dust concentrations on the 25th April? If so are the authors happy the dust model wasn't still spinning up on May 5th when first profile was evaluated?

Overall, when reading the model descriptions I wasn't clear which model simulated only dust versus all aerosols. Also which models include meteorological data assimilation or not. This is important for dust transport and vertical mixing. Also across all models what forecast range was being evaluated?

I also think a bit more detail on the dust schemes themselves used in each model would be useful across all 3 models.
P9 L33/P10 L1-2 : This description of how the model variability was calculated was initially not clear to me. Please make it clear that you took 8 different model profiles at gridpoints surrounding the ships location as well as the profile matching the lidar location if this is the correct interpretation.

P10 L28 – why was NMMB data not available for case 1? Is this related to the spin-up issue I question above or just purely technical?

P10 L33 again needs to be clearer here that you use the METEOSAT imagery to assess the impact of wet deposition on the retrieved lidar profiles.

P11 L9 : Most models should at the very least conserve mass therefore I wouldn't expect numerical losses referred to here – suggest removing this statement

Section 5 Conclusions: A succinct summary of the key findings of the paper would be useful here. What can modellers learn / take away from this study?

P13 L33 "mass concentration deviated partly strongly" ?? I don't think you can deviate partly and strongly at the same time!

I would recommend that the authors include a comment on the potential of lidar data for assimilation into aerosol models to complement the AOD assimilation currently employed. The authors should have the expertise to comment on the usefulness of such data. I note they comment on the use of Deep Blue to further improve models. **ACPD**

---

## Author Comment (AC1) · 31 Oct 2017

The comment was uploaded in the form of a supplement:
https://www.atmos-chem-phys-discuss.net/acp-2017-502/acp-2017-502-AC1-supplement.pdf

---

## Author Response (AR1)

Dear Editor, Dear Reviewers!

Thank you for careful reading of the manuscript and for the numerous suggestions, which are widely considered and triggered further fruitful discussion.

Let us start with an overview of the main changes:

We increased the number of authors, now in addition: Oriol Jorba (NMMB/BSC-Dust) and Samuel Remy (MACC/CAMS). We replaced Angela Benedetti by Samual Remy.

We updated Figures 6 and 7 (we added missing NMMB/BSC-Dust forecasts of 23 May 2013).

We added a new figure (Figure 9): comparison of dust fine mode fraction: observations (lidar, AERONET) vs simulation (MACC/CAMS, NMMB/BSC-Dust).

The paper of Mamouri and Ansmann (AMT, 2017) is now published (it was cited as 'in preparation', although already available as AMTD version to that time). In this paper, it is outlined how fine dust and coarse dust mass concentration and extinction coefficient can be obtained from lidar observations.

We extended the discussion on dust size distribution during emission based on Kok et al. (2011) and Mahowald et al. (2014). The brittle fragmentation theory is in agreement with observations and predicts a fine dust fraction of 1% during emission, whereas the saltation theory predicts 5-10% relative fine dust contribution. Thus the models (based on saltation theory) assume much higher fine dust emissions as obviously observed. This is now stated and the consequences are discussed.

Step-by-step answers to the comments of the reviewers:

Our answers in bold. The changed text parts in the revised manuscript are also given in bold.

Reviewer #1

- The writing can be improved in places. In particular, at the beginning of sections 2 and 3 a brief explanation of what is to follow in that section and how that builds towards addressing the main objective would be helpful. Related to that, reiterating the paper's main objectives in those key locations would help focus the reader.

This is done! We follow these suggestions. New paragraphs are written (Sect. 2 and 3).

- I cannot comment on the validity of the lidar retrievals or the POLIPHON method, as this is outside of my expertise. This is quite critical to the paper, so I hope another reviewer on this paper or the companion paper can comment on this.

The Mamouri and Ansmann (2017) paper is meanwhile reviewed and published as AMT paper (also as a contribution to the SALTRACE special issue as the two R/V Meteor papers).

- The constants that convert measured backscatter coefficients to light extinction coefficients are given as single numbers. However, the light extinction results reported in this paper are quite sensitive to these conversion constants, and so it's important that the authors provide realistic

uncertainties on these numbers (and propagate them) so that their results can be interpreted correspondingly.

**Done! Section 2, page 4, last paragraph. The lidar ratio uncertainty is about 10 sr, and thus 20% when using the dust lidar ratio of 55sr.**

- P. 5, l. 18-22: The description here of how the extinction-to-volume conversion factors are obtained is a bit problematic, because the cited paper (Mamouri and Ansmann, 2017) is in prep and not accessible. So more explanation should be given here. Does this conversion account for dust asphericity or does it assume spherical particles? Also, references should be included for the assumed particle densities.

**As mentioned, the Mamouri and Ansmann (2017) paper is meanwhile reviewed and published. 'Everything' is described there in rather large detail. So, we do not see the need to expand the discussion here significantly. Some more info is given on page 6.**

- P. 5, l. 25-26: The reader is referred here again to an in prep manuscript (Mamouri and Ansmann, 2017) for details on the methodology to separate fine and coarse dust, so I'd suggest replacing this reference with the published Mamouri and Ansmann (2014). Since this is a key component of the methodology, a more detailed review of this method would be very helpful to the reader. It's unclear to me how the different depolarization ratios allow a decomposition of the signal into fine dust, coarse dust, and non-dust. That still seems like an underconstraint problem, so more explanation is necessary here.

**We agree and provide more information (page 6). But the lidar method works well as the consistency between our lidar-derived fine dust fraction and the one obtained from AERONET photometer observations indicate. However, note that AERONET provides a column-integrated value, which is influenced by all particles, … dust, marine and smoke particles…. To corroborate the reliability of the fine dust profiles obtained with the polarization lidar (a very different approach to the complementary AERONET approach)…. we add the new Figure 9.**

- P. 5, l.28-30: Assuming a height-independent size distribution is an important simplification that requires experimental support. Is this shown by other SALTRACE measurements? Some references are needed here.

**This is fully justified as the SALTRACE observations indicate. The lidar profiles of the depolarization ratio and of the lidar ratio were usually height independent (Haarig et al, 2017, Barbados observations) and the aircraft observations of microphysical properties in the Barbados area during SALTRACE 2013 showed the same height-independent features (Weinzierl et al., 2017). From this we can conclude that the size distribution was almost the same from SAL base to top.**

- P. 5, l.30-31: the authors here claim "full agreement" with shipborne fine and coarse dust AOT. For this statement to be convincing, it should be shown, especially as the authors use this to further assert that their lidar observations "reflect very well the true fine-to-coarse dust extinction and mass conditions." Otherwise, these statements should be removed.

**Yes, that motivated us to present the new Figure 9! …which shows nicely the consistency between AERONET and lidar dat, and in contrast … the overestimation of the fine-to-coarse dust ratio by the models.**

- Section 3: Since the simulated fine and coarse dust abundances are compared against measurements in Figs. 7 and 8, the authors should provide the emitted fine and coarse fractions for each model. This helps the reader interpret whether the model discrepancies are due to errors in emission, transport, or deposition.

**Done! The information is given in Sect.3, and repeated in Sect.4, when we discuss the overestimation of the fine dust fraction during long-range transport in detail…. We state that the overestimated emitted fine dust fraction seems to be one reason for the overestimated fine dust fraction after long range transport over the Atlantic.**

- P. 13, l. 23-5: I think the authors have enough information to at least hypothesize about the reasons for the model disagreement. For instance, the discrepancy close to source regions, where model errors in transport and deposition are minimal, suggests that models have a problem with their emitted size distribution. The fact that this discrepancy increases with distance from the source region suggests that dust in models is depositing too fast, which the authors already alluded to in other places in the manuscript. I consider these two of the paper's main take-home messages, so I would suggest the authors sum that up here.

**This is now discussed in detail in Sect. 4, and summarized in the conclusion section 5.**

……………………………………………………………………………………………………………………………………………………………………..

**Reviewer #2**

Overall the manuscript is well written and clear. There are typos which I haven't listed here (can do so if required) so would recommend a thorough checking through of paper for such errors and mistakes prior to publishing.

**We carefully read the revised version several times and … ACP will help us to remove final typos and mistakes (during the copyediting and proofreading process).**

P5 L23-29: It would be nice to see the lidar retrievals independently verified if possible. The authors refer to agreement with shipborne sun photometer observations. I would recommend this verification is shown, particularly as the relevant reference Mamouri and Ansmann, 2017 is still in prep.

**The Mamouri and Ansmann (2017) paper is meanwhile published as AMT paper (as a contribution to the SALTRACE special issue as the two R/V Meteor papers). We add a new Figure 9 (Sect. 4) and compare the lidar-derieved dust fine mode fraction with AERONET observations (and the NMMB and MACC model results). We also started to compare Falcon aircraft in situ observations of the fine mode fractions with the ones obtained with lidar at Barbados during SALTRACE in the summer of 2013. We found very good agreement, and we work on a publication.**

P6, Section 3.1: Some extra detail is needed in terms of the model description of SKIRON. What region does the SKIRON model cover? What resolution is the model run at – this is important in terms of the skill of the model in simulating the dust emission which is dependent on wind speed. Does the model have its own meteorological data assimilation to constrain the meteorological variables or is it just free-running? Again this is important information in terms of assessing the dust transport. Please include some more information plus a bit more detail on the dust model itself.

Is SKIRON an operational dust forecasting model and therefore dust products were available or was a dedicated experiment conducted, this is not clear from the description here.

**We follow the suggestion of the reviewer and expanded the description of SKIRON accordingly and answer the questions of the reviewer in Sect. 3.1 (pages 7-8).**

P7, L2-3 and L5: The MACC/CAMS simulation tool –> poor choice of wording , would suggest changing tool –>system. The key feature of the MACC/CAMS system is that Earth Observation data are operationally assimilated to provide Near-real time forecasts for a wide range of forecast products – this wasn't clear to me in the opening sentences of Section 3.2 – suggest making it clearer.

**Done (see Sect. 3.2, pages 8-9).**

P7 L18: My understanding of the MACC system is that they don't yet assimilate Deep Blue retrievals over bright surfaces (ie: not that they were not available as stated in the text) – I would suggest ensuring your statements here are accurate and again the key point being that MACC do not assimilate AOD over bright desert surfaces.

**As mentioned we expanded the discussion in Sect.3.2 on data assimilation in the MACC/CAMS system.**

P7 L20: Again the key point with the MACC aerosol assimilation is that while it is very good at constraining the total AOT, the control variable in the assimilation is the total aerosol mass mixing ratio and the increment to the total mixing ratio needs to be then redistributed across all individual aerosol species which is done based on the model background. So while the total AOD should be well constrained the speciation and therefore dust contribution to the total extinction is not. This is clear on P10 L5-10. I think these key features of the MACC/CAMS system need to be made more clear for the reader.

**Yes, this is stated (two times, Sect 3.2 and Sect.4.1). The total AOT is well constrained, but no attempt is made to adjust the specific contributions of marine, dust and smoke particles.**

P7 L24-26: Two different model resolutions are mentioned here which is confusing. Are the authors saying the model simulation is run at 0.8x0.8 deg but output products used in the present analysis were only available at 1.125 deg? Clarification required.

**The simulations are done with a fixed resolution. However, as a user, you can go to the MACC/CAMS data base and download your products for the resolution of your choice. We selected 1.125 deg (page 9).**

P8 L15-18 The MACC model also contributes to WMO-SDS WAS and ICAP (I'm not sure about SKIRON) so authors should be consistent in descriptions or remove the statements. Appropriate reference for ICAP is Sessions et al: https://www.atmoschem-phys.net/15/335/2015/acp-15-335-2015.html

**We added this information in Sect. 3.3 (page 10).**

P9 L2: Was the model initiated from 0 dust concentrations on the 25th April? If so are the authors happy the dust model wasn't still spinning up on May 5th when first profile was evaluated?

**Cold start at 28 April 2013… is now mentioned in Sect 3.3 (final paragraph).**

Overall, when reading the model descriptions I wasn't clear which model simulated only dust versus all aerosols. Also which models include meteorological data assimilation or not. This is important for dust transport and vertical mixing. Also across all models what forecast range was being evaluated?

**All this is now added as much as possible (Sect. 3.1, 3.2, 3.3).**

I also think a bit more detail on the dust schemes themselves used in each model would be useful across all 3 models.

**We provide a bit more information, but we believe that more detailed explanations of the dust schemes are not needed as long as we provide proper references…. Therefore, we state more often: More details can be found in references X, Y, Z.**

P9 L33/P10 L1-2 : This description of how the model variability was calculated was initially not clear to me. Please make it clear that you took 8 different model profiles at gridpoints surrounding the ships location as well as the profile matching the lidar location if this is the correct interpretation.

**This is written now more clearly in Sect. 3.1 (SKIRON) and later on in Sect. 4.1 (to check the variability in the MACC profiles).**

P10 L28 – why was NMMB data not available for case 1? Is this related to the spin-up issue I question above or just purely technical?

**We (TROPOS, lidar team) simply made a mistake and ordered just NMMB simulations up to the 22 May 2013, …. four years ago… So, we had to run the Barcelona model again. Now the results for 23 May 2013 are included, and presented in Figs. 6 and 7.**

P10 L33 again needs to be clearer here that you use the METEOSAT imagery to assess the impact of wet deposition on the retrieved lidar profiles.

**Improved! (Sect. 4.2, pages 12-13)**

P11 L9 : Most models should at the very least conserve mass therefore I wouldn't expect numerical losses referred to here – suggest removing this statement

**But we are still convinced that numerical diffusion is a problem that should at least not be ignored… We mention it in the last paragraph in Sect. 4.**

Section 5 Conclusions: A succinct summary of the key findings of the paper would be useful here. What can modellers learn / take away from this study?

**We agree and follow widely the suggestion of the reviewer. We asked all groups: What did you learn, what are new points now…? And then we rewrote the entire conclusion section. We believe, now the updated conclusion section is greatly improved and more interesting to read… and hopefully meets the recommendation of the reviewer.**

P13 L33 "mass concentration deviated partly strongly" ?? I don't think you can deviate partly and strongly at the same time!

**Agree! ..is removed!**

I would recommend that the authors include a comment on the potential of lidar data for assimilation into aerosol models to complement the AOD assimilation currently employed. The authors should have the expertise to comment on the usefulness of such data. I note they comment on the use of Deep Blue to further improve models.

**Done (Sect. 5, final paragraph)**

[revised manuscript text omitted]